# Evaluation of Serum Neurofilament Light Chain and Glial Fibrillary Acidic Protein as Screening and Monitoring Biomarkers for Brain Metastases

**DOI:** 10.3390/cancers13092227

**Published:** 2021-05-06

**Authors:** Su-Hyun Kim, Ho-Shin Gwak, Youngjoo Lee, Na-Young Park, Mira Han, Yeseul Kim, So-Yeon Kim, Ho Jin Kim

**Affiliations:** 1Department of Neurology, National Cancer Center, Goyang 10408, Korea; sua177@ncc.re.kr (N.-Y.P.); 74034@ncc.re.kr (Y.K.); 75249@ncc.re.kr (S.-Y.K.); hojinkim@ncc.re.kr (H.J.K.); 2Department of Cancer Control, National Cancer Center Graduate School of Cancer Science and Policy, Goyang 10408, Korea; nsghs@ncc.re.kr; 3Department of Internal Medicine, National Cancer Center, Goyang 10408, Korea; yjlee@ncc.re.kr; 4Biometric Research Branch, National Cancer Center, Goyang 10408, Korea; hmr0209@ncc.re.kr

**Keywords:** brain metastasis, neurofilament light chain, glial fibrillary acidic protein, blood-based biomarker

## Abstract

**Simple Summary:**

Approximately 20% of patients with cancer develop brain metastases (BM). Early BM diagnosis is critical to enable less invasive or toxic approaches. Sensitive and easy-to-use blood-based BM biomarkers may allow early diagnosis and appropriate timely treatment and may improve overall survival. This study aimed to evaluate the potential roles of serum neurofilament light chain (sNfL) and serum glial fibrillary acidic protein (sGFAP) for diagnosing and monitoring BM. We found significant differences in the sNfL and the sGFAP levels between patients with and without BMs. The optimal cutoff-levels of sNfL and sGFAP had sensitivities of 91% and 91%, respectively, and combining the two biomarkers (sNfL or sGFAP) improved the sensitivity to up to 98%, with an overall accuracy higher than 91%. Thus, sNfL and sGFAP may be used as biomarkers for BM screening in patients with cancer.

**Abstract:**

We evaluated the potential serum neurofilament light chain (sNfL) and serum glial fibrillary acidic protein (sGFAP) roles in diagnosing and monitoring brain metastases (BMs). We included 70 patients with newly diagnosed BMs, 71 age- and cancer type-matched patients without BMs, and 67 healthy controls (HCs). We compared sNfL and sGFAP levels among the groups using a single-molecule array immunoassay. We prospectively followed 26 patients with BMs every 2–3 months by measuring sNfL and sGFAP levels and performing magnetic resonance imaging (MRI) scans. The sNfL and the sGFAP levels were higher in patients with BMs (medians: sNfL, 63.7 pg/mL; sGFAP, 819.5 pg/mL) than in those without BMs (sNfL, 13.3 pg/mL; sGFAP, 154 pg/mL; *p* < 0.001) and HCs (sNfL, 12.5 pg/mL; sGFAP, 135 pg/mL; *p* < 0.001). The sNfL and the sGFAP cutoff levels had a sensitivity and a specificity of 91%. The sGFAP cutoff level had a sensitivity of 91% and a specificity of 97%. The sNfL and the sGFAP levels were related to the BM size but not to the primary cancer type. After BM treatment, sNfL and sGFAP levels decreased with reduced BM lesions on MRI; however, they increased when BMs progressed. sNfL and sGFAP are potential biomarkers for BM screening in cancer patients.

## 1. Introduction

Approximately 20% of patients with cancer develop brain metastases (BMs) [1,2]. The incidence of BMs is increasing with increased surveillance, improved imaging technologies, systemic control, and prolonged survival [3]. Early BM diagnosis is critical to enable less invasive or less toxic approaches, such as stereotactic radiosurgery, rather than whole-brain radiotherapy (RT) or surgery [3,4,5]. 

Magnetic resonance imaging (MRI) is the most important diagnostic tool for evaluating the presence of BMs [3]. It is usually the first test performed when BM symptoms, such as headaches and seizures, are observed. However, BMs are less frequently symptomatic than expected: only 19% of patients with newly diagnosed BMs have neurological symptoms [6]. Currently, routine surveillance brain MRI screening is not implemented for the majority of cancers in asymptomatic patients. It is recommended for those with newly diagnosed lung cancer and stage III–IV melanoma, but there is still heterogeneity across countries in real practice, presumably because of limited access to MRI; only 63% of patients with stage III lung cancer are screened [7]. Furthermore, approximately 50% of patients with locally advanced lung cancer and 10–30% of patients with metastatic breast cancer develop BMs [3,4,8,9], but there are no specific guidelines on monitoring the development of BMs. As MRI is costly and requires a long time to perform an examination, routine MRI surveillance at frequent intervals in all patients with cancer is not feasible. In this scenario, the identification of easily accessible blood-based BM biomarkers could allow early diagnosis and appropriate timely treatment and may improve overall survival.

The neurofilament light chain (NfL) is a cytoskeletal protein expressed in large caliber myelinated axons. Increased NfL levels in peripheral blood were found in several diseases characterized by axonal loss in the central (CNS) or the peripheral nervous system, including multiple sclerosis, stroke, dementia, and chemotherapy-induced peripheral neuropathy [10,11,12,13,14]. The glial fibrillary acid protein (GFAP) is an intermediate filament highly expressed in astrocytes and serves as a marker of astrocyte activation/injury [15]. The recently emerged single molecule array (SIMOA) testing, an ultrasensitive enzyme-linked immunosorbent assay technique with a sensitivity approximately 1000-fold higher than that of traditional ligand binding assays, was used to assess blood NfL and GFAP levels, leading to the renaissance of these biomarkers in neurological diseases [10,16,17]. The objective of this study was to evaluate the potential roles of the serum NfL and GFAP levels in screening and monitoring of BMs as single and combined biomarkers.

## 2. Results

### 2.1. Patients

Seventy patients with BMs, 71 patients with solid cancers without BMs, and 67 healthy controls (HCs) were enrolled (Table 1). No differences in age or sex were observed between patients with and without BMs and HCs. Forty-nine patients with BMs (70%) had neurological symptoms or signs, including mental change, hemiparesis, facial palsy, dysarthria, ataxia, headache, or dizziness, while 21 (30%) were asymptomatic.

### 2.2. Comparison of the Serum NfL (sNfl) and GFAP (sGFAP) Levels in Patients with and without BMs and HCs

The sNfL and the sGFAP levels increased with age in HCs (*r* = 0.572 and *r* = 0.578, respectively, both *p* < 0.001). In a previous study, we reported that oxaliplatin increased the sNfL but not the sGFAP levels [11]; therefore, we compared the data of patients with and without recent cytotoxic chemotherapy (within 6 months before sampling, including platinum- or taxol-based agents, or irinotecan). Among patients without BMs, those undergoing cytotoxic chemotherapy (*n* = 11) showed higher sNfL levels (median 62.6 pg/mL (interquartile range (IQR), 40.7, 154)) than those without undergoing recent chemotherapy (13.3 pg/mL (IQR 10.0, 19.9), *p* < 0.001) and HCs (12.7 (IQR, 9.8, 18.6), *p* < 0.001), but no differences in the sGFAP levels among the groups were noted. Likewise, among patients with BMs, those who underwent recent cytotoxic chemotherapy (*n* = 15) showed higher sNfL levels than those without undergoing chemotherapy (162.5 pg/mL (68.6, 218.8) vs. 68.8 pg/mL (IQR 37.9, 133), respectively, *p* = 0.021), but no differences in the sGFAP levels were observed. Therefore, we excluded patients who underwent recent chemotherapy from further sNfL analysis. The sNfL and the sGFAP levels in each group are presented in Figure 1. Patients with BMs had higher median sNfL and sGFAP levels (63.7 pg/mL (IQR, 29.9, 133) and 819.5 pg/mL (IQR, 321, 1650), respectively) than those without BMs (13.3 pg/mL (IQR, 10.1, 19.9) and 154 pg/mL (IQR, 121, 189), respectively) or HCs (12.5 pg/mL (IQR, 9.8, 17.9) and 135 pg/mL (IQR, 90.7, 185), respectively) (all *p* < 0.001), while no significant differences in sNfL or sGFAP levels between patients without BMs and HCs were noted.

### 2.3. sNfL and sGFAP Level as BM Diagnostic Markers

The receiver operating characteristic (ROC) analysis provided optimal cutoff points for differentiating between patients with and without BMs in each age category (Table 2). The sNfL cutoff level at each age category had a sensitivity of 91% and a specificity of 91%, while the sGFAP cutoff level had a sensitivity of 91% and a specificity of 97% (Table 3). Combined analysis using sNfL or sGFAP levels at their respective cutoffs showed a sensitivity of 98% and a specificity of 88%. In patients with asymptomatic BMs, sNfL and sGFAP showed sensitivities of 81% and 86%, respectively, while the combined analysis of the two biomarkers (sNfL or sGFAP) had a sensitivity of 94% (75% when using both sNfL and sGFAP).

All patients without BM were of stage I–II, while two-thirds of patients with BM had significant extracranial tumor volume. Of the 70 patients with BM, 23 had solitary BM without extracranial tumors including the primary site on positron emission tomography–computed tomography (CT), chest CT, and/or abdominopelvic CT. To exclude the possibility that sNfL and sGFAP are simply related with overall tumor volume including BM rather than just BM lesions, we compared sNfL and sGFAP levels between the 23 patients with solitary BM and the other 47 patients with BM and extracranial tumor. Patients with solitary BM had a younger mean age (mean, 53 years (standard deviation (SD), 10) vs. 63 years (SD, 9); *p* = 0.001) than those with BM and extracranial diseases, and there was no significant difference in the lesion size of BM between the two groups (mean, 3.5 cm (SD, 2.0) vs. 4.2 cm (SD, 1.8); *p* = 0.21). When we compared the serum NfL and GFAP levels after adjusting for age, there was no significant difference in the sNfL (median, 51.4 pg/mL (IQR, 28.35, 100) vs. 79.7 pg/mL (IQR, 40.9, 141.5); *p* = 0.252) and sGFAP levels (median, 661 pg/mL (IQR, 262, 1640) vs. 873 pg/mL (IQR, 326, 2081); *p* = 0.766) between those with solitary BM and those with BM and extracranial diseases.

### 2.4. Correlations of sNFL and sGFAP Levels with Clinical and MRI Features

In patients with BMs, no significant differences in sNfL or sGFAP levels were observed among different types of primary cancers. Symptomatic patients (*n* = 49) with BMs showed higher median sGFAP levels compared to asymptomatic patients (*n* = 21) with BMs (1086 pg/mL (IQR, 376, 2572.5) vs. 337 pg/mL (IQR, 239.5, 764.5), respectively; *p* = 0.015), but no significant differences were observed in the sNfL levels (*p* = 0.057). The median sum of diameters of target lesions was 3.5 cm (IQR, 2.6, 5.5). The lesion size showed a significant positive correlation with sNFL (*r* = 0.621, *p* < 0.001) and sGFAP levels (*r* = 0.49, *p* < 0.001) after adjusting for age (Figure 2). Of the 70 patients with BMs, five had false-negative results on the sNfL test (single meta (2.6–2.8 cm) in the cerebellum (*n* = 4) or in the cerebrum (*n* = 1)) and five on the sGFAP test (single meta (0.9–1.7 cm) in the cerebellum (*n* = 3) or in the cerebrum (*n* = 2)), while only one had false-negative results on both tests (three lesions (≤0.6 cm) in the frontal lobe) (Figure 3).

### 2.5. Longitudinal Follow-Up

We followed up sNfL and sGFAP levels and the MRI status of all 26 patients at 2–3 month intervals for a median period of 9 months (IQR, 6, 10). The median age at BM diagnosis was 68 years (IQR, 58, 71). Lung cancer (73%) was the most frequent type, followed by breast (15%) and other cancers (12%). Twelve, nine, three, and two patients underwent surgeries followed by RT and/or chemotherapy, surgeries and/or chemotherapy, and only chemotherapy, respectively. During the follow-up period, 19 patients showed complete (CR) or partial response (PR) for a median period of 6 months followed by progression, two revealed PR followed by RT-induced necrosis, and five showed CR. At BM diagnosis, three, two, and one patient had false-negative results for sNfL, sGFAP, and for both, respectively. All these patients showed progression during the follow-up period, and the sNfL and sGFAP levels increased above the cutoff levels (for BM diagnosis) at BM progression. After BM treatment, sNfL and sGFAP levels decreased (Figure 4), in line with the reduction of BM lesions on MRI. The sNFL (median, 33.4 pg/mL vs. 81.7 pg/mL, *p* = 0.037) and the sGFAP levels (median, 238 pg/mL vs. 701 pg/mL, *p* < 0.001) in the last sample obtained before BM progression were significantly lower than those in the sample obtained at the initial BM diagnosis. However, sNfL and sGFAP levels never decreased below the cutoff levels for BM diagnosis in 16 (62%) patients, including five patients who exhibited CR at 6 months after BM treatment. In 19 patients with BM progression, at the time of progression, sNfL (median, 75.4 pg/mL vs. 45.9 pg/mL, *p* < 0.001) and sGFAP levels (median, 567 pg/mL vs. 334 pg/mL, *p* < 0.001) were higher than those observed 2–3 months before (Figure 4C,D and Appendix A). During the follow-up period after BM diagnosis, nine patients received cytotoxic chemotherapy for a median period of 4.5 months; chemotherapy increased the sNfL levels, but not the sGFAP levels, independently of BM progression (Figure 4B). Two patients with RT-induced necrosis during the follow-up period also showed increased sNfL and sGFAP levels (Figure 4E).

## 3. Discussion

The present study found significant differences in sNfL and sGFAP levels between patients with and without BMs. The optimal cutoff levels of sNfL and sGFAP estimated per age group had sensitivities of 91% and 91%, respectively, and combining the two biomarkers (sNfL or sGFAP) improved the sensitivity to up to 98%, with an overall accuracy higher than 91%. Among asymptomatic patients, the combination of sNfL and sGFAP also showed a high sensitivity of 94%. The sNfL and the sGFAP levels were related to the BM size but not to the type of primary cancer. Longitudinal changes in these levels were associated with changes in BMs. Thus, sNfL and sGFAP may be used as biomarkers for BM screening in patients with cancer, regardless of the primary tumor, and potentially for the early detection of BM recurrence.

The development of sensitive, reproducible, and easy-to-use biomarkers to detect BMs is highly desirable. Blood is easy to access and allows repeated measurements. Some studies suggested that serum S100B can be a biomarker for the early detection of BMs [18,19,20,21], but these results were not supported by other studies [22,23]. With the availability of an ultra-high sensitivity NfL and GFAP assay, it became feasible to monitor damages to neurons or astrocytes through blood analysis. Neuroaxonal injury induced by inflammation in the tumor microenvironment and metastatic growth of invading cancer cells may lead to increased sNfL in patients with BMs. Astrogliosis was reported around BMs in human brain tissue and in animal models [24,25]. The upregulation of GFAP following reactive astrogliosis and destruction of astrocyte structural integrity combined with blood–brain barrier disruption may lead to increased sGFAP levels in patients with BMs [26,27]. Although an increasing number of studies reported the roles of sNFL and sGFAP in various neurological disorders, data are scarce for patients with BMs [28,29]. Hepner et al. showed higher concentrations of sNfL and sGFAP in 22 patients with gliomas or BMs compared to those in nine patients without BMs [28]. Another recent study showed higher sNfL levels in 43 patients with BMs compared to those observed in 25 without BMs, although there was no age-matching in the comparison [29]. Interestingly, increased NfL levels were found at a median period of 3 months before BM diagnosis; the possibility of asymptomatic BMs at 3 months before BM diagnosis was also not negligible, as no MRI scans were conducted at that time [29]. In our study, 95% of patients with asymptomatic BMs also revealed sNfL or sGFAP levels above the respective cutoffs. Patients with BMs who had false-negative results for sNfL or sGFAP tended to show a small lesion or lesions limited to the cerebellum. In this study, it was not determined whether sNfL and sGFAP levels increased before BMs were observed on MRI.

After BM treatment, sNfL and sGFAP levels decreased compared with those at the initial BM diagnosis, in line with the reduction of BM lesions on MRI. These results suggested that treatments could control brain injury from BMs and that sNfL and sGFAP may reflect the ongoing CNS injury in real time. Consistently, when BMs progressed, significant sNfL and sGFAP increases were noted. Nevertheless, the detection of BM progression in longitudinal monitoring had several limitations. First, sNFL and sGFAP levels remained well above the respective cutoffs in more than half of the patients. Such continuous release of NfL and GFAP from the brain tissue into the bloodstream is attributable to the loss of integrity in axonal neurofilaments and astrocyte structure in the necrotic tumor tissue and to radiation-induced neurotoxicity and blood–brain barrier disintegration within the BMs [26]. Thus, it is difficult to determine whether BMs recurred using the cutoff levels. Second, increases in sNfL and sGFAP levels are not BM-specific, but they reflect the magnitude and perhaps the tempo of neuroaxonal and astrocytic damage in any pathology. Any CNS injury, such as RT-induced necrosis or stroke, would also increase sNfL and sGFAP levels, thus making it difficult to differentiate BM recurrence from other phenomena with these markers alone. Furthermore, while GFAP was found to be highly CNS specific, as relevant extracerebral sources of this protein are not identified, NfL can be elevated because of peripheral nerve axonal injury [11]. Thus, the value of monitoring the sNfL levels for BM recurrence in patients undergoing cytotoxic chemotherapy is limited. Finally, these biomarkers are quantitative measurements associated with lesion burden. While large BM lesions decrease after treatment, recurrence of surrounding small lesions may not induce significant increases of these proteins, thus escaping detection.

A general limitation of this study was the lack of an independent validation cohort. All patients with lung cancer without BMs were confirmed to be BM-free on MRI examination, but not those with breast or other cancers, who, however, had no evidence of disease for 5 years. In addition, caution should be given to the threshold reported for sNfL and sGFAP, as studies identifying the optimal cutoff levels may be subject to selection bias and may not be replicable [30]. As we included only patients without history of neurological disorders and diabetes mellitus, the proposed cutoff levels for early BM detection might be low positive predictive values, especially in those with other underlying neurological diseases or diabetes mellitus. Finally, since these two biomarkers are not released exclusively from the brain, the damage to the peripheral nervous system or the spinal cord by extensive extracranial tumor invasion may contribute to some extent to the increase of these biomarkers in some patients with BM. Comparing sNfL and sGFAP levels in patients with BM and without BM but with extensive extracranial metastases can help clarify that these biomarkers are more specific to BM rather than metastases in other tissues.

## 4. Materials and Methods

### 4.1. Study Design and Participants

This cross-sectional and longitudinal cohort study included 70 patients with pathologically confirmed lung, breast, and other solid cancers with newly diagnosed BMs on MRI and/or biopsy between January 2015 and November 2020 at the National Cancer Center, Korea. The blood samples and data of 34 patients with BMs were provided by the Bio Bank of the National Cancer Center (NCC), Korea, and 36 consecutive patients with BMs were recruited in this study for the prospective follow-up and cross-sectional analyses (Figure 5).

Age- and stage I–II primary cancer-matched patients with lung, breast, and other cancers without BMs and age-matched HCs were included in the control group. Blood samples and data of patients with cancers but without BMs and HCs who underwent general health checkups were provided by the NCC Bio Bank. Patients and HCs with diabetes mellitus or a clinical diagnosis of neurological disorders including stroke, neurodegenerative diseases, or peripheral nerve diseases were excluded. Among the 36 patients with BMs who enrolled for prospective follow-up examinations, 10 were excluded because of short follow-up periods (<6 months). Thus, 26 patients with BMs were prospectively checked by measuring the serum NfL (sNfL) and GFAP (sGFAP) levels and performing brain MRI scans at every 2–3-month intervals to identify the recurrence of BMs.

For patients with newly diagnosed BMs, blood samples were obtained within 1 month after MRI and before BM treatment. Brain MRI examinations were performed as part of routine care at initial staging, during regular follow-ups, and in cases of clinical symptoms leading to suspected BMs. For patients with cancer without BMs, blood samples were obtained just before surgery for the primary tumor. All patients with lung cancer without BMs were confirmed to be without BMs according to the MRI examination findings at the time of blood sampling. Patients with breast, colon, or ovary cancer without BMs had negative brain MRI reports at initial staging or during follow-up examinations or had completed a clinical follow-up examination period of at least 5 years with no signs of recurrent or metastatic disease.

Clinical data, including age, sex, primary tumor, history of cancer treatment, presenting symptoms at BM diagnosis, and radiological BM features (sum of the longest diameters of up to five target lesions when more than one lesion was present) [22], were collected from medical records. In the longitudinal cohort, treatment histories after BM diagnosis and BM outcomes on MRI were also collected. The radiographic response and progression of BM were assessed using the neuro-oncology criteria: progression was defined when the sum of linear measurements exceeded 20% over the baseline or new lesions were identified; CR was defined as the disappearance of all CNS target lesions for at least 4 weeks, with no new lesions; PR was defined as the reduction in the sum of linear measurements by ≥30% compared with baseline [31].

### 4.2. sNfL and sGFAP Measurement

Blood samples were obtained via venous puncture and centrifuged immediately for 10 min at 2000× *g*. Serum was transferred into Eppendorf tubes and stored at −80 °C until measurement. The analyses were performed using an in-house assay on the SIMOA platform (Quanterix, Lexington, MA, USA), as per the manufacturer’s instructions. The intra-assay coefficient of variation of the assay was <10%. Laboratory analyses were blinded to the patient’s identity or diagnosis.

### 4.3. Statistical Analyses

To compare clinical characteristics among the groups, Kruskal–Wallis and Fisher’s exact tests were performed for continuous and categorical variables, respectively. Differences in sNfL and sGFAP levels among the groups were analyzed using Kruskal–Wallis and Bonferroni post-hoc tests. Correlations between the serum protein levels and age were analyzed using Spearman’s correlation analysis. Partial correlation analyses were performed to test whether the maximal size of BM lesions contributed to sNfL and sGFAP levels after adjusting for age. The sNfL and the sGFAP levels were compared between serial assessments using the Wilcoxon signed-rank test. ROC curve analysis was used to determine optimal cutoff values. The sNfL and the sGFAP cutoff levels for BM diagnosis were analyzed by building age categories in a 10 year range. The related diagnostic accuracies were evaluated by computing sensitivity, specificity, positive and negative predictive values, and accuracy. Values of *p* < 0.05 were considered statistically significant. Analyses were performed using GraphPad PRISM and STATA, version 15 (StataCorp LP, College Station, TX, USA).

## 5. Conclusions

In summary, the sNfL and the sGFAP levels may be used as novel biomarkers for early BM detection. Although their specificity for BMs is not high, these blood-based biomarkers can easily provide information regarding the CNS injuries. Thus, we consider sNfL and sGFAP as screening tools for the development of occult BMs but not as standalone diagnostic tools. The use of sNfL and sGFAP levels as biomarkers to select patients who would need brain MRI examination could impact the cost of initial staging. In particular, these biomarkers can be helpful in patients with advanced cancer who need close surveillance for BM development. Furthermore, sNfL and sGFAP might be potential biomarkers for the early detection of BM recurrence. Validation studies in larger cohorts of patients are needed to confirm the accuracy and the consistency of sNfL and sGFAP as early screening and monitoring markers for BMs.

## Figures and Tables

**Figure 1 cancers-13-02227-f001:**
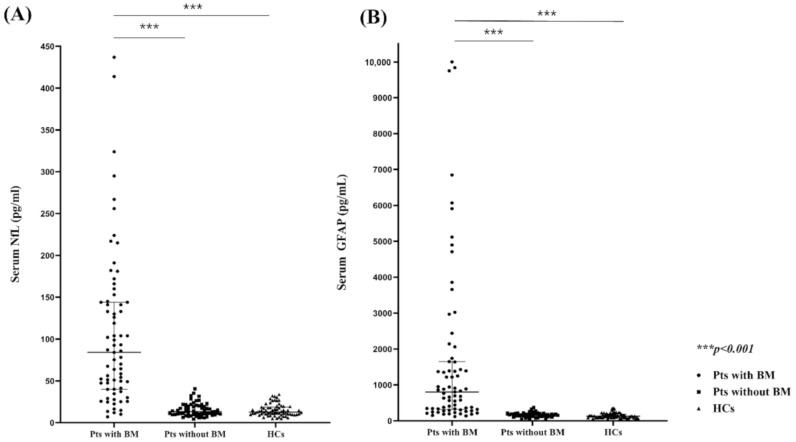
Comparison of the serum NfL (**A**) and GFAP levels (**B**) between patients with BMs, patients without BMs, and healthy controls. NfL, neurofilament light chain, GFAP, glial fibrillary acidic protein; BMs, brain metastases.

**Figure 2 cancers-13-02227-f002:**
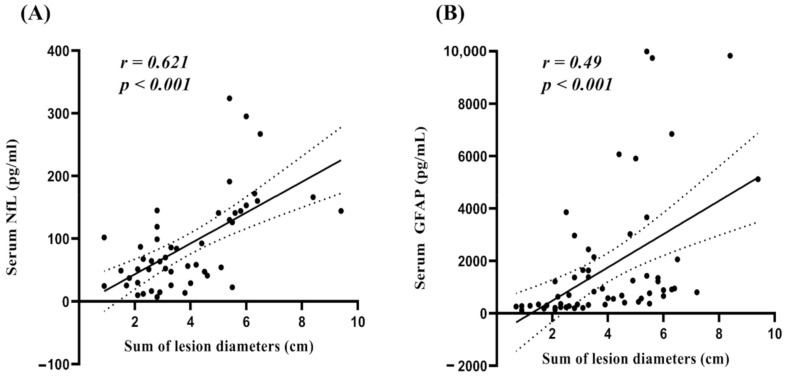
Age-adjusted partial correlation between the diameter of lesion and sNfL (**A**) and sGFAP levels (**B**). The scatterplot shows regression lines and 95% confidence intervals. sNfL, serum neurofilament light chain; sGFAP, serum glial fibrillary acidic protein.

**Figure 3 cancers-13-02227-f003:**
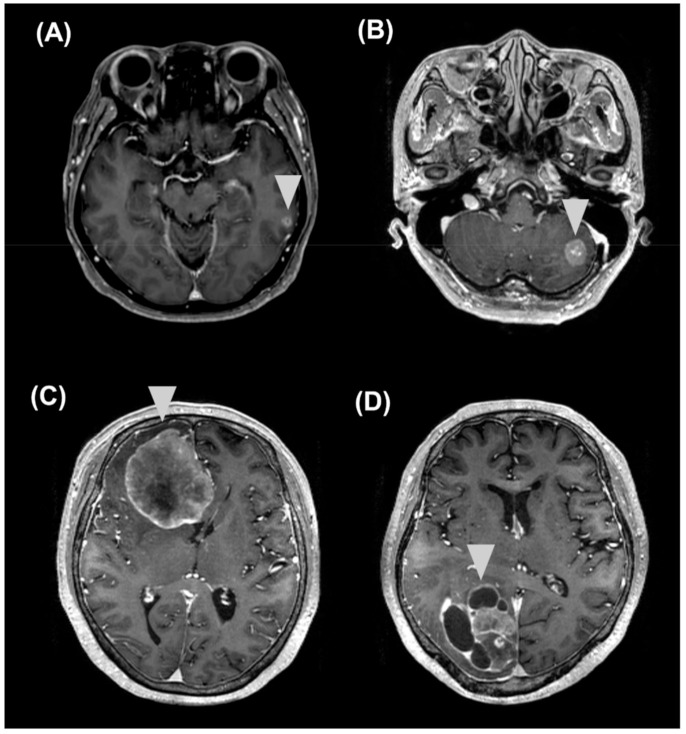
Representative examples of MRI findings in patients with BMs. (**A**,**B**) Findings of patients with false-negative results in the sNfL or sGFAP test: (**A**) Findings of a 71-year-old patient with lung cancer and three small brain metastatic lesions (longest diameter < 0.5 cm; sNfL, 25.5 pg/mL; sGFAP, 185 pg/mL) and (**B**) those of a 47-year-old patient with breast cancer and a single cerebellar lesion (longest diameter 1.6 cm; sNfL, 34.6 pg/mL; sGFAP, 150 pg/mL). (**C**,**D**) Findings of patients with much higher sNfL and sGFAP levels than those of patients with BMs: (**C**) findings of a 62-year-old patient with breast cancer and a huge single brain metastatic lesion (longest diameter 6 cm) in the right frontal lobe (sNfL, 130 pg/mL; sGFAP, 10,000 pg/mL) and (**D**) those of a 73-year-old patient with lung cancer and a large multiseptated brain metastatic lesion (longest diameter 6 cm) in the right occipital lobe (sNfL, 172 pg/mL; sGFAP, 6849 pg/mL). The triangle indicates the BM lesion. BM, brain metastasis; sNfL, serum neurofilament light chain; sGFAP, serum glial fibrillary acidic protein; MRI, magnetic resonance imaging.

**Figure 4 cancers-13-02227-f004:**
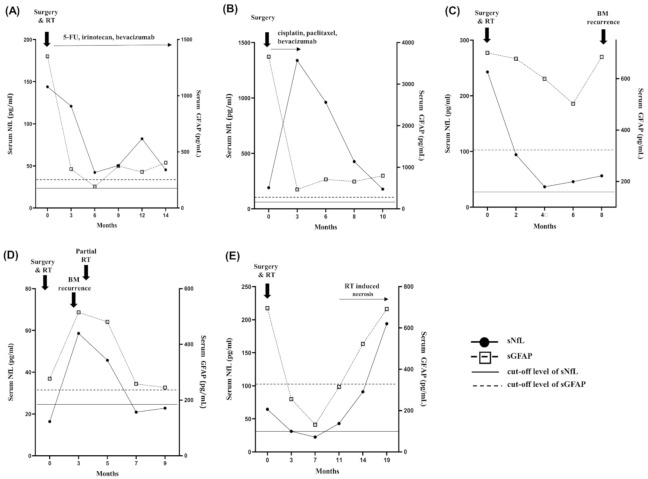
Longitudinal patterns of sNfL and sGFAP levels after BM diagnosis in five representative patients. (**A**) After BM treatment, sGFAP and sNfL levels decreased significantly, then, sGFAP level remained at a low level during the follow-up period without BM recurrence, and sNfL levels slightly increased throughout chemotherapy treatment (**B**) After BM treatment, sGFAP levels decreased significantly, and remained at a low level during the follow-up period without BM recurrence, while sNfL levels increased rapidly due to the effects of chemotherapy and gradually decreased after stopping chemotherapy (**C**) After BM treatment, sGFAP and sNfL levels decreased but increased by BM recurrence. (**D**) Even after BM treatment, sNfL and sGFAP levels increased significantly by BM recurrence and decreased in line with the reduction of BM lesions after additional RT (**E**) After BM treatment, sNfL and sGFAP levels decreased but both increased significantly by RT-induced necrosis. The longitudinal follow-up results of sNFL and sGFAP levels of the other 17 patients with BM progression are shown in the Appendix A. sNfL, serum neurofilament light chain; sGFAP, serum glial fibrillary acidic protein; BM, brain metastasis; RT, radiotherapy.

**Figure 5 cancers-13-02227-f005:**
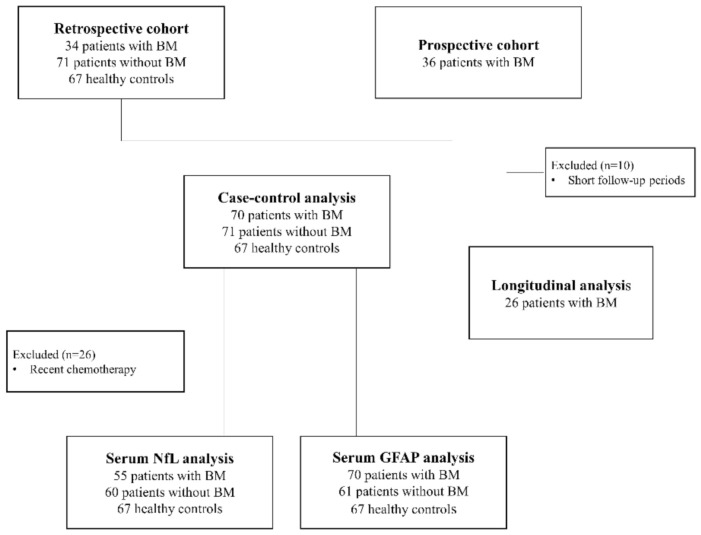
Flow diagram for the enrollment of study participants. sNfL, serum neuro filament light chain; sGFAP, serum glial fibrillary acidic protein; BM, brain metastasis.

**Table 1 cancers-13-02227-t001:** Clinical characteristics of the study population.

Clinical Characteristics	Patients with BMs (*n* = 70)	Patients without BMs (*n* = 71)	Healthy Controls (*n* = 67)	*p*
Age, years, median (IQR)	60 (51–69)	60 (54–59)	60 (54–69)	0.953
Sex, female (%)	69	56	71	0.162
Primary cancer, (*n*)	Lung cancer (39) Breast cancer (26) Colon cancer (2) Ovary cancer (2) Cervical cancer (1)	Lung cancer (34) Breast cancer (30) Colon cancer (3) Ovary cancer (2) Cervical cancer (2)	N/A	0.797
Time from initial cancer diagnosis to BMs, months, median (IQR)	22.5 (8–50)	N/A	N/A	N/A

BMs, brain metastases; IQR, interquartile range; N/A, not applicable.

**Table 2 cancers-13-02227-t002:** The cutoff levels of sNfL and sGFAP for differentiating patients with and without BMs in each age group. Levels above the cutoff level in the blood test were deemed positive while those equal to or below the cutoff level were deemed negative.

sNfL			sGFAP		
	AUC	Cut-Off (pg/mL)		AUC	Cut-Off (pg/mL)
31–50 years (*n* = 41)	0.88	11.8	31–50 years (*n* = 44)	0.986	185
51–60 years (*n* = 49)	1.0	21.8	51–60 years (*n* = 58)	0.961	189
61–70 years (*n* = 63)	0.936	23.9	61–70 years (*n* = 72)	0.964	246
71–80 years (*n* = 29)	0.958	35.2	71–80 years (*n* = 34)	0.939	371

BMs, brain metastases; sNfL, serum neurofilament light chain; sGFAP, serum glial fibrillary acidic protein; AUC, area under the curve.

**Table 3 cancers-13-02227-t003:** Sensitivity, specificity, predictive values, and accuracy of sNfL and sGFAP levels in BM detection.

Protein	Sensitivity, % (95% CI)	Specificity, % (95% CI)	PPV, % (95% CI)	NPV, % (95% CI)	Accuracy, % (95% CI)
sNfL	91 (0.8–0.96)	91 (0.84–0.95)	81 (0.68–0.89)	95 (0.91–0.99)	91 (0.85–0.94)
sGFAP	91 (0.82–0.97)	97 (0.93–0.99)	94 (0.86–0.98)	96 (0.91–0.98)	95 (0.91–0.98)
sNfL or sGFAP	98 (0.90–0.99)	88 (0.81–0.93)	78 (0.67–0.87)	99 (0.95–0.99)	91 (0.82–0.95)
sNfL and sGFAP	87 (0.75–0.95)	99 (0.96–0.99)	98 (0.89–0.99)	95 (0.92–0.98)	96 (0.92–0.98)

BM, brain metastasis; PPV, positive predictive value; NPV, negative predictive value; sNfL, serum neurofilament light chain; sGFAP, serum glial fibrillary acidic protein; CI, confidence interval.

## Data Availability

The datasets generated during and/or analyzed during the current study are available from the corresponding author on reasonable request.

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
