# Peer review of "Evaluation of Serum Neurofilament Light Chain and Glial Fibrillary Acidic Protein as Screening and Monitoring Biomarkers for Brain Metastases"

_cancers, 2021, doi:10.3390/cancers13092227_

Round 1
Reviewer 1 Report
Kim et al. have performed a well-written manuscript on a very interesting topic: early diagnosis of brain metastases (BM) in cancer patients. Here, they present a study on serum NfL and GFAP - proteins expressed in glial cells and myelinated axons – evaluated in cancer patients with and without BM as well as healthy cases. They find that NfL and GFAP, independently of cancer type, can be applied as diagnostic markers of BM individually, however, the performance of the markers improve significantly by combining the two. Furthermore, in a few selected patients, they demonstrate that longitudinal monitoring of these biomarkers can hold a clinical relevance as the biomarker levels increase with BM progression.
The overall impression of the manuscript is good, however, I do have some concerns that need attendance
- It is not clear which patients are included in which part of the study. The authors describe that they include a total of 70 patients with BM, 71 without BM and 67 healthy controls and basic clinical characteristics are presented for these patients in table 1. Afterwards, the authors describe that patients who had received chemotherapy within the last 6 months prior to blood sampling were excluded as chemotherapy could affect the level of NfL. How many patients are then included in the study? Further, in- and exclusion criteria must be stated clearly in the Methods section. Was NfL and GFAP measureable in all blood samples? According to REMARK, when presenting a biomarker study, a flowchart is recommended to help the reader to follow the selection of patients.
- The authors apply ROC-analyses to define the optimal cut-point for each biomarker thereby introducing data-dependent cut points and not predefined cut point (this is not in line with REMARK recommendations). Applying data-dependent cut points increases the risk of finding a positive association and by combining to the two markers NfL and GFAP with each their data-dependent cut-point the risk of positive overestimation is unavoidable. Are predefined cut points available? Or are the identified cut points in line with other studies? If predefined cut-points are not available, the authors should add a section about the pitfalls of this approach in the discussion.
- The authors address the age-dependent NfL level but it is unclear how many patients with and without BM are in each age interval.
4. Longitudinal follow-up. A total of 26 patients were included in this part of the study of which 19 had CR or PR in 6 months followed by progression. In the manuscript 5 patients have been selected for graphic demonstration. It could be interesting for the readers to see the graphics of the remaining 14 patients. Are there a clear pattern or is the full picture a bit more blurry (could be applied as a supplementary figure)
Author Response
Reviewer 1
Kim et al. have performed a well-written manuscript on a very interesting topic: early diagnosis of brain metastases (BM) in cancer patients. Here, they present a study on serum NfL and GFAP - proteins expressed in glial cells and myelinated axons – evaluated in cancer patients with and without BM as well as healthy cases. They find that NfL and GFAP, independently of cancer type, can be applied as diagnostic markers of BM individually, however, the performance of the markers improve significantly by combining the two. Furthermore, in a few selected patients, they demonstrate that longitudinal monitoring of these biomarkers can hold a clinical relevance as the biomarker levels increase with BM progression. The overall impression of the manuscript is good, however, I do have some concerns that need attendance
Response: The authors would like to thank the reviewer for his/her constructive critique to improve the manuscript. We have made every effort to address the issues raised and to respond to all comments. Please, find next a detailed, point-by-point response to the reviewer's comments.
It is not clear which patients are included in which part of the study. The authors describe that they include a total of 70 patients with BM, 71 without BM and 67 healthy controls and basic clinical characteristics are presented for these patients in table 1. Afterwards, the authors describe that patients who had received chemotherapy within the last 6 months prior to blood sampling were excluded as chemotherapy could affect the level of NfL. How many patients are then included in the study? Further, in- and exclusion criteria must be stated clearly in the Methods section. Was NfL and GFAP measureable in all blood samples? According to REMARK, when presenting a biomarker study, a flowchart is recommended to help the reader to follow the selection of patients.
Response: We would like to apologize to the reviewer for the confusion. Please note that we enrolled a total of 70, 71, and 67 patients with BM, patients without BM, and healthy controls, respectively. As the chemotherapy had no effect on the sGFAP levels and affected only the sNfL levels, the sNfL levels of 21 patients (15 and 11 patients with and without BM, respectively) who underwent chemotherapy were excluded from the comparison. Following the reviewer’s suggestion, we have added a flowchart describing the patients’ selection and the performed procedures (Figure 5).
The authors apply ROC-analyses to define the optimal cut-point for each biomarker thereby introducing data-dependent cut points and not predefined cut point (this is not in line with REMARK recommendations). Applying data-dependent cut points increases the risk of finding a positive association and by combining to the two markers NfL and GFAP with each their data-dependent cut-point the risk of positive overestimation is unavoidable. Are predefined cut points available? Or are the identified cut points in line with other studies? If predefined cut-points are not available, the authors should add a section about the pitfalls of this approach in the discussion.
Response: As the studies that focused on these two biomarkers using SIMOA are recent, the data regarding the normal value observed in the healthy control group using the same method are limited. We agree with the reviewer’s comment. Following the reviewer’s suggestion, we have added the following sentence to the limitation part of the revised manuscript: “In addition, caution should be given to the threshold reported for sNfL and sGFAP, as studies identifying the optimal cutoff levels may be subject to selection bias and may not be replicable.” (Lines 260–262).
The authors address the age-dependent NfL level but it is unclear how many patients with and without BM are in each age interval.
Response: Following the reviewer’s suggestion, we have presented the number of patients of each age group in Table 2.
Longitudinal follow-up. A total of 26 patients were included in this part of the study of which 19 had CR or PR in 6 months followed by progression. In the manuscript 5 patients have been selected for graphic demonstration. It could be interesting for the readers to see the graphics of the remaining 14 patients. Are there a clear pattern or is the full picture a bit more blurry (could be applied as a supplementary figure)
Response: In Figure 4, two patients with BM progression were included. Following the reviewer’s suggestion, we have presented the longitudinal follow-up results of the other 17 patients with BM progression during the longitudinal follow-up period in a supplementary figure S1.

Reviewer 2 Report
The authors propose sNfL and sGFAP as biomarkers for Brain Metastasis. From the data, there is no doubt that these two markers represent some kind of disease activity, and I have the impression that the data is interesting.
On the other hand, I would be cautious to conclude from this data that sNFL and sGFAP are specific biomarkers of BM. First of all, as the authors show, the concentration of these two factors in serum seems to have a parallel relationship with tumor volume. In patients with BM, the overall tumor volume is usually higher than in patients without BM, so the possibility that these two factors are simply markers of stage progression and overall tumor volume is not excluded. In other words, to propose that the two factors are brain metastasis-specific markers, the following data are required.
1. Higher serum levels in patients with BM compared to patients with the same tumor volume. In other words, no significant difference in tumor volume between Pts with BM and without BM.
2. Higher serum concentrations in patients with BM compared to patients with other metastatic sites (e.g., bone or liver metastases).
However, even if these data do not come out positively, I think these two factors would be important as a normal tumor marker, so the article would worth publishing. However, I think the hypothesis that it is a brain metastasis-specific marker will be rejected, so I think you should change the claim of your paper.
Author Response
Reviewer 2
The authors propose sNfL and sGFAP as biomarkers for Brain Metastasis. From the data, there is no doubt that these two markers represent some kind of disease activity, and I have the impression that the data is interesting.
Response: The authors would like to thank the reviewer for his/her constructive critique to improve the manuscript. We have made every effort to address the issues raised and to respond to all comments. Please, find next a detailed, point-by-point response to the reviewer's comments.
On the other hand, I would be cautious to conclude from this data that sNFL and sGFAP are specific biomarkers of BM. First of all, as the authors show, the concentration of these two factors in serum seems to have a parallel relationship with tumor volume. In patients with BM, the overall tumor volume is usually higher than in patients without BM, so the possibility that these two factors are simply markers of stage progression and overall tumor volume is not excluded. In other words, to propose that the two factors are brain metastasis-specific markers, the following data are required. Higher serum levels in patients with BM compared to patients with the same tumor volume. In other words, no significant difference in tumor volume between Pts with BM and without BM.
Response: We would like to thank the reviewer for the comment. In this study, we enrolled stage I–II patients without BM as a control group. Nevertheless, approximately two-thirds of patients with BM had a significant extracranial tumor volume. Thus, the suggested comparison could not be performed. Nevertheless, although patients without BM had a certain extracranial tumor volume, there was no difference in the serum NfL and GFAP levels from those of healthy controls, suggesting that these markers are not simply related to the overall tumor volume. Additionally, we conducted the following analysis to show that the serum NfL and GFAP levels were not simply related to the overall tumor volume but were originated from BM lesions. Of the 70 patients with BM, 23 had solitary BM without extracranial tumors including the primary site on positron emission tomography–computed tomography (CT), chest CT, and/or abdominopelvic CT. We compared the two biomarkers between the 23 patients with solitary BM and other 47 patients with BM and extracranial tumor. Patients with solitary BM had a younger mean age (mean, 53 years [standard deviation [SD], 10] vs. 63 years [SD, 9]; p=.001) than those with BM and extracranial diseases, and there was no significant difference in the lesion size of BM between the two groups (mean, 3.5 cm [SD, 2.0] vs. 4.2 cm [SD, 1.8]; p=.21). When we compared the serum NfL and GFAP levels after adjusting for age, there was no significant difference in the sNfL (median, 51.4 pg/µL [IQR, 28.35, 100] vs. 79.7 pg/µL [IQR, 40.9, 141.5]; p=.252) and sGFAP levels (median, 661 pg/µL [IQR, 262, 1640) vs. 873 pg/µL (IQR, 326, 2081); p=.252) between those with solitary BM and those with BM and extracranial diseases. Based on the aforementioned findings, it seems that the elevation of the sNfL and sGFAP levels was caused by BM lesions rather than by the overall tumor volume. We have provided this information in the revised manuscript as follows:
“All patients without BM were of stage I–II, while two-thirds of patients with BM had significant extracranial tumor volume. Of the 70 patients with BM, 23 had solitray BM without extracranial tumors including the primary site on positron emission tomography–computed tomography (CT), chest CT, and/or abdominopelvic CT. To exclude the possibility, sNfL and sGFAP are simply related with overall tumor volume including BM, rather than just BM lesions, we compared the sNfL and sGFAP levels between the 23 patients with solitary BM and other 47 patients with BM and extracranial tumor. Patients with solitary BM had a younger mean age (mean, 53 years [standard deviation [SD], 10] vs. 63 years [SD, 9]; p=.001) than those with BM and extracranial diseases, and there was no significant difference in the lesion size of BM between the two groups (mean, 3.5 cm [SD, 2.0] vs. 4.2 cm [SD, 1.8]; p=.21). When we compared the serum NfL and GFAP levels after adjusting for age, there was no significant difference in the sNfL (median, 51.4 pg/µL [IQR, 28.35, 100] vs. 79.7 pg/µL [IQR, 40.9, 141.5]; p=.252) and sGFAP levels (median, 661 pg/µL [IQR, 262, 1640) vs. 873 pg/µL (IQR, 326, 2081); p=.252) between those with solitary BM and those with BM and extracranial diseases.” (Lines 127–141)
Higher serum concentrations in patients with BM compared to patients with other metastatic sites (e.g., bone or liver metastases). Response: There was no patient with distant metastasis among those without BM. Thus, this comparison was not possible. However, through the analysis presented above, we have shown that these two markers originate from lesions of BM and not from metastasis of other tissues.

Round 2
Reviewer 2 Report
I think this is a very valuable paper, and thank you for the additional validation.
However, it is impossible to determine from the additional validation that sNfL and sGFAP are specific for brain metastasis.
The authors compared patients with solitary BM to those with BM and extracranial diseases and found no significant difference, which is a corroboration that they are specific for brain metastasis.
In particular, the median value was higher for BM and extracranial disease, so without a figure, we cannot judge whether the argument is being properly made or not. In addition, this additional validation is small, and I wonder whether the hypothesis that "tumor volume is related to the blood concentration of these two factors" can be rejected just because the P value is greater than 0.05. I think it would be better to mention this point in Limitation.
Author Response
Response: The authors would like to thank the reviewer for his/her constructive critique to improve the manuscript. Here we attach the figure of the comparison results of sNfL and sGFAP between patients with solitary BM and patients with BM and extracranial tumor.
Following the reviewer’s suggestion, we have added the following sentences to the limitation part of the revised manuscript: “Finally, since these two biomarkers are not released exclusively from the brain, the damage to the peripheral nervous system or spinal cord by extensive extracranial tumor invasion may have contributed to some extent to the increase of these biomarkers in some patients with BM. Comparing sNfL and sGFAP levels in patients with BM and without BM but with extensive extracranial metastases can help clarify that these biomarkers are more specific for BM rather than metastases in other tissues (Lines 265-270).
